# Learning Others' Intentional Models in Multi-Agent Settings Using Interactive POMDPs

**Yanlin Han    Piotr Gmytrasiewicz**
Department of Computer Science
University of Illinois at Chicago
Chicago, IL 60607
{yhan37,piotr}@uic.edu

## Abstract

Interactive partially observable Markov decision processes (I-POMDPs) provide a principled framework for planning and acting in a partially observable, stochastic and multi-agent environment. It extends POMDPs to multi-agent settings by including models of other agents in the state space and forming a hierarchical belief structure. In order to predict other agents' actions using I-POMDPs, we propose an approach that effectively uses Bayesian inference and sequential Monte Carlo sampling to learn others' intentional models which ascribe to them beliefs, preferences and rationality in action selection. Empirical results show that our algorithm accurately learns models of the other agent and has superior performance than methods that use subintentional models. Our approach serves as a generalized Bayesian learning algorithm that learns other agents' beliefs, strategy levels, and transition, observation and reward functions. It also effectively mitigates the belief space complexity due to the nested belief hierarchy.

## 1   Introduction

Partially observable Markov decision processes (POMDPs) [11] is a general decision-theoretic framework for planning under uncertainty in a partially observable, stochastic environment. An autonomous agent acts rationally in such settings by constantly maintaining beliefs of the physical state and sequentially choosing the optimal actions that maximize the expected value of future rewards. Thus, solutions of POMDPs map an agent's beliefs to actions. Although POMDPs can be used in multi-agent settings, it usually treats the impacts of other agents' actions as noise and embeds them into the state transition function. Examples of such POMDPs are Utile Suffix Memory [14], infinite generalized policy representation [13], and infinite POMDPs [3]. Therefore, an agent's beliefs about other agents are not part of the solutions of POMDPs.

Interactive POMDPs (I-POMDPs) [7] generalize POMDPs to multi-agent settings by replacing POMDP belief spaces with interactive belief systems. Specifically, an I-POMDP augments the plain beliefs about the physical states in POMDP by including models of other agents. The models of other agents included in the new augmented belief space consist of two types: intentional models and subintentional models. An intentional model ascribes beliefs, preferences, and rationality to other agents [7], while a simpler subintentional model, such as finite state controllers [15], does not. The augmentation of intentional models forms a hierarchical belief structure that represents an agent's belief about the physical state, belief about the other agents and their beliefs about others' beliefs, and so on. Solutions of I-POMDPs map an agent's belief about the environment and other agents' models to actions. It has been shown [7] that the added sophistication of modeling others as rational agents results in a higher value function compared to one obtained from treating others as noise, which implies the modeling superiority of I-POMDPs over other approaches.

However, the interactive belief augmentation of I-POMDPs results in a drastic increase of the belief space complexity, because the agent models grow exponentially as the belief nesting level increases. Therefore, the complexity of the belief representation is proportional to belief dimensions, which is known as the curse of dimensionality. Moreover, due to the fact that exact solutions to POMDPs are PSPACE-complete and undecidable for finite and infinite time horizon respectively[16], the time complexity of more generalized I-POMDPs is at least PSPACE-complete and undecidable for finite and infinite horizon, since an I-POMDP may contain multiple POMDPs or I-POMDPs of other agents. Due to these complexities, a solution which accounts for an agent's belief over an entire intentional model has not been implemented up to date. There are partial solutions that depend on what is known about other agents' beliefs about the physical states [2], but they do not include the state of an agent's knowledge about others' reward, transition, and observation functions. Indirect approach such as subintentional finite state controllers [15] do not include any of these elements either. To unleash the full modeling power of intentional models and mitigate the aforementioned complexities, a robust approximation algorithm is needed. The purpose of this algorithm is to compute the nested interactive belief over elements of the intentional models and predict other agents' actions. It is crucial to the trade-off between solution quality and computational complexity.

To address these issues, we propose an approach that uses Bayesian inference and customized sequential Monte Carlo sampling [4] to obtain approximate solutions to I-POMDPs. We assume that the modeling agent maintains beliefs over intentional models of other agents and make sequential Bayesian updates using observations from the environment. While in multi-agent settings, others agents' models other than their beliefs are usually assumed to be known, in our assumption the modeling agent does not know any information about others' beliefs, strategy levels, and transition, observation, and reward functions. It only relies on learning indirectly from observations about the environment, which is influenced by others agents' actions. Since this Bayesian inference task is analytically intractable due to the requirement of computing high dimensional integrations, we have devised a customized sequential Monte Carlo method, extending the interactive particle filter (I-PF) [2] to the entire intentional model space. The main idea of this method is to descend the nested belief hierarchy, parametrize other agents' model functions, and sample all model parameters at each nesting level according to observations.

Our approach successfully recovers other agents' models over the intentional model space which contains their beliefs, strategy levels, and transition, observation and reward functions. It extends I-POMDP's belief update to larger model space, and therefore it serves as a generalized Bayesian learning method for multi-agent systems in which other agents' beliefs, transition, observation and reward functions are unknown. By approximating Bayesian inference using a customized sequential Monte Carlo sampling method, we significantly mitigate the belief space complexity of I-POMDPs.

## 2 The Model

### 2.1 I-POMDP framework

I-POMDPs [7] generalize POMDPs [11] to multi-agent settings by including models of other agents in the belief state space. The resulting hierarchical belief structure represents an agent's belief about the physical state, belief about the other agents and their beliefs about others' beliefs, and can be nested infinitely in this recursive manner. Here we focus on the computable counterparts of infinitely nested I-POMDPs: finitely nested I-POMDPs. For simplicity of presentation, we consider two interacting agents $i$ and $j$. This formalism generalizes to more number of agents in a straightforward manner.

A finitely nested interactive POMDP of agent $i$ , I-POMDP$_{i,l}$, is defined as:

$$I\text{-}POMDP_{i,l} = \langle IS_{i,l}, A, \Omega_i, T_i, O_i, R_i \rangle \tag{1}$$

where $IS_{i,l}$ is a set of interactive states, defined as $IS_{i,l} = S \times M_{j,l-1}, l \geq 1$, $S$ is the set of physical states, $M_{j,l-1}$ is the set of possible models of agent $j$, and $l$ is the strategy (nesting) level. The set of models, $M_{j,l-1}$, can be divided into two classes, the intentional models, $IM_{j,l-1}$, and subintentional models, $SM_j$. Thus, $M_{j,l-1} = IM_{j,l-1} \cup SM_j$.

The *intentional* models, $IM_{j,l-1}$, ascribe beliefs, preferences, and rationality in action selection to other agents, thus they are analogous to *types*, $\theta_j$, used in Bayesian games [10]. The *intentional* models, $\Theta_{j,l-1}$, of agent $j$ at level $l-1$ is defined as $\theta_{j,l-1} = \langle b_{j,l-1}, A, \Omega_j, T_j, O_j, R_j, OC_j \rangle$,

where $b_{j,l-1}$ is agent $j$'s belief nested to the level $(l-1)$, $b_{j,l-1} \in \Delta(IS_{j,l-1})$, and $OC_j$ is $j$'s optimality criterion. It can be rewritten as $\theta_{j,l-1} = \langle b_{j,l-1}, \hat{\theta}_j \rangle$, where $\hat{\theta}_j$ includes all elements of the intentional model other than the belief and is called the agent $j$'s frame.

The *subintentional* models, $SM_j$, constitute the remaining models in $M_{j,l-1}$. Examples of subintentional models are finite state controllers [15], no-information models [8] and fictitious play models [5].

The $IS_{i,l}$ can be defined in an inductive manner:

$$
\begin{aligned}
& IS_{i,0} = S, && \Theta_{j,0} = \{\langle b_{j,0}, \hat{\theta}_j \rangle : b_{j,0} \in \Delta(IS_{j,0})\} && M_{j,0} = \Theta_{j,0} \cup SM_j \\
& IS_{i,1} = S \times M_{j,0}, && \Theta_{j,1} = \{\langle b_{j,1}, \hat{\theta}_j \rangle : b_{j,1} \in \Delta(IS_{j,1})\} && M_{j,1} = \Theta_{j,1} \cup M_{j,0} \\
& \quad\ldots\ldots \\
& IS_{i,l} = S \times M_{j,l-1}, && \Theta_{j,l} = \{\langle b_{j,l}, \hat{\theta}_j \rangle : b_{j,l} \in \Delta(IS_{j,l})\} && M_{j,l} = \Theta_{j,l} \cup M_{j,l-1}
\end{aligned}
\tag{2}
$$

All remaining components in an I-POMDP are similar to those in a POMDP. $A = A_i \times A_j$ is the set of joint actions of all agents. $\Omega_i$ is the set of agent i's possible observations. $T_i : S \times A \times S \to [0,1]$ is the transition function. $O_i : S \times A \times \Omega_i \to [0,1]$ is the observation function. $R_i : IS_i \times A \to \mathbb{R}$ is the reward function.

## 2.2 Interactive belief update

Given the definitions above, the interactive belief update can be performed as follows, by considering others' actions and anticipated observations:

$$
\begin{aligned}
b_{i,l}^t(is^t) &= Pr(is^t | b_{i,l}^{t-1}, a_i^{t-1}, o_i^t) \\
&= \alpha \sum_{is^{t-1}} b_{i,l}(is^{t-1}) \sum_{a_j^{t-1}} Pr(a_j^{t-1} | \theta_{j,l-1}^{t-1}) T(s^{t-1}, a^{t-1}, s^t) O_i(s^t, a^{t-1}, o_i^t) \\
&\quad \times \sum_{o_j^t} O_j(s^t, a^{t-1}, o_j^t) \tau(b_{j,l-1}^{t-1}, a_j^{t-1}, o_j^t, b_{j,l-1}^t)
\end{aligned}
\tag{3}
$$

Compared with POMDP, the interactive belief update in I-POMDP takes two additional elements into account. First, the probability of other's actions given his models needs to be computed since the state now depends on both agents' actions (the second summation). Second, the modeling agent needs to update his beliefs based on the anticipation of what observations the other agent might get and how it updates (the third summation).

Similarly to POMDPs, the value associated with a belief state in I-POMDPs can be updated using value iteration:

$$
V(\theta_{i,l}) = \max_{a_i \in A_i} \left\{ \sum_{is \in IS} b_{i,l}(is) ER_i(is, a_i) + \gamma \sum_{o_i \in \Omega_i} P(o_i | a_i, b_{i,l}) V(\langle SE_{\theta_i}(b_{i,l}, a_i, o_i), \hat{\theta}_i \rangle) \right\}
\tag{4}
$$

where $ER_i(is, a_i) = \sum_{a_j} R_i(is, a_i, a_j) Pr(a_j | \theta_{j,l-1})$.

Then the optimal action, $a_i^*$, for an infinite horizon criterion with discounting, is part of the set of optimal actions, $OPT(\theta_i)$, for the belief state:

$$
OPT(\theta_{i,l}) = \arg\max_{a_i \in A_i} \left\{ \sum_{is \in IS} b_{i,l}(is) ER_i(is, a_i) + \gamma \sum_{o_i \in \Omega_i} P(o_i | a_i, b_{i,l}) V(\langle SE_{\theta_i}(b_{i,l}, a_i, o_i), \hat{\theta}_i \rangle) \right\}
\tag{5}
$$

## 3 Sampling Algorithms

The Markov Chain Monte Carlo (MCMC) method [6] is widely used to approximate probability distributions that are difficult to compute directly. Sequential versions of Monte Carlo methods, such as as particle filters [1], work on sequential inference tasks, especially sequential decision making under Markov assumption. At each time step, a particle filter draws samples (or particles) from a

proposal distribution, commonly the conditional distribution $p(x_t|x_{t-1})$ of the current state $x_t$ given the previous $x_{t-1}$, then uses the observation function $p(y_t|x_t)$ to compute importance weights for all particles and resample them according to the weights.

The Interactive Particle Filter (I-PF) was devised as a filtering algorithm for interactive belief update in I-POMDP, which generalizes the classic particle filter algorithm to multi-agent settings [2]. It uses the state transition function as the proposal distribution, which is usually used in a specific particle filter algorithm called bootstrap filter [9]. However, due to the enormous belief space, the I-PF implementation assumes that the other agent's frame $\hat{\theta}_j$ is known to the modeling agent, therefore it simplifies the belief update from $S \times \Theta_{j,l-1}$ to a significantly smaller space $S \times \{b_{j,l-1}\}$, where $j$ represents the other agent and $\Theta_{j,l-1}$ is $j$'s model space.

Our interactive belief update described in Algorithm 1 and 2, however, generalizes I-POMDP's belief update to larger intentional model space which contains other agents' beliefs, and transition, observation and reward functions. In the remaining part of this section, we will firstly give a brief introduction of our algorithms and discuss the motivations of each sampling step. Then we will show the major differences between our algorithm and the I-PF, since this generalization is nontrivial. A concrete example of the algorithm is given in Figure 1 in the next section as well.

---

**Algorithm 1: Interactive Belief Update**

---

$\tilde{b}_{k,l}^{t} = \text{InteractiveBeliefUpdate}(\tilde{b}_{k,l}^{t-1}, a_k^{t-1}, o_k^t, l > 0)$

1    For $is_k^{(n),t-1} = < s^{(n),t-1}, \theta_{-k,l-1}^{(n),t-1} > \in \tilde{b}_{k,l}^{t-1}$:

2        sample $a_{-k}^{t-1} \sim P(a_{-k}|\theta_{-k,l-1}^{(n),t-1})$

3        sample $s^{(n),t} \sim T_k(s^t|s^{(n),t-1}, a_k^{t-1}, a_{-k}^{t-1})$

4        for $o_{-k}^t \in \Omega_{-k}$:

5           if $l = 1$:

6               $b_{-k,0}^{(n),t} = \text{Level0BeliefUpdate}(\theta_{-k,0}^{(n),t-1}, a_{-k}^{t-1}, o_{-k}^t)$

7               $is_k^{(n),t} = < s^{(n),t}, \theta_{-k,0}^{(n),t} >$

8           else:

9               $b_{-k,l-1}^{(n),t} = \text{InteractiveBeliefUpdate}(\tilde{b}_{-k,l-1}^{(n),t-1}, a_{-k}^{t-1}, o_{-k}^t, l-1)$

10             $\theta_{-k,l-1}^{(n),t} = < b_{-k,l-1}^{(n),t}, \hat{\theta}_{-k,l-1}^{(n),t-1} >$

11             $is_k^{(n),t} = < s^{(n),t}, \theta_{-k,l-1}^{(n),t} >$

12             $w_t^{(n)} = O_{-k}^{(n)}(o_{-k}^t|s^{(n),t}, a_k^{t-1}, a_{-k}^{t-1})$

13             $w_t^{(n)} = w_t^{(n)} \times O_k(o_k^t|s^{(n),t}, a_k^{t-1}, a_{-k}^{t-1})$

14             $\tilde{b}_{k,l}^{temp} = < is_k^{(n),t}, w_t^{(n)} >$

15    normalize all $w_t^{(n)}$ so that $\sum_{n=1}^{N} w_t^{(n)} = 1$

16    resample $\{is_k^{(n),t}\}$ from $\tilde{b}_{k,l}^{temp}$ according to normalized $\{w_t^{(n)}\}$

17    resample $\theta_{-k,l-1}^{(n),t} \sim N(\theta_{-k,l-1}^t|\theta_{-k,l-1}^{(n),t-1}, \Sigma)$

18    return $\tilde{b}_{k,l}^t = is_k^{(n),t} = < s^{(n),t}, \theta_{-k,l-1}^{(n),t} >$

---

The Algorithm 1 requires inputs of the modeling agent's prior belief, $\tilde{b}_{k,l}^{t-1}$, which is represented as a set of $n$ samples $is_k^{(n),t-1}$, along with the action, $a_k^{t-1}$, the observation, $o_k^t$, and the belief nesting level, $l > 0$. Here $k$ represents either agent $i$ or $j$, and $-k$ represents the other agent, $j$ or $i$, correspondingly. We assume that the modeled agent's action set $A_{-k}$, observation set $\Omega_{-k}$ and optimality criteria $OC_k$ are known to all agents. We want to learn the other agent's initial belief about the physical state, $b_{-k}^0$, the transition function, $T_{-k}$, the observation function, $O_{-k}$ and the reward function, $R_{-k}$.

The initial belief samples, $is_k^{(n),t-1}$, are generated from the prior nested belief in the similar way as described in the I-PF literature [2] except that $T_{-k}^{(n)}, O_{-k}^{(n)}$, and $R_{-k}^{(n)}$ are sampled from their prior distributions as well. Notice that $T_{-k}^{(n)}, O_{-k}^{(n)}$, and $R_{-k}^{(n)}$ are all part of the frame, namely $\hat{\theta}_{-k}^{(n)} = < A_{-k}, \Omega_{-k}, T_{-k}^{(n)}, O_{-k}^{(n)}, R_{-k}^{(n)}, OC_k >$, as appeared in line 7 and 11 in Algorithm 1.

With initial belief samples, the Algorithm 1 starts from propagating each sample forward in time and computing their weights (line 1-15), then it resamples according to the weights and similarity between models (line 16-18). Intuitively, the samples associated with actual observations perceived by agent $k$ will gradually carry larger weights and be resampled more often, therefore they will approximately represent the exact belief. Specifically, for each of $is_k^{(n),t-1}$, the algorithm samples the other agent's optimal actions $a_{-k}^{t-1}$ given its model from $P(A_{-k}|\theta_{-k}^{(n),t-1})$ (line 2), which equals $\frac{1}{|OPT|}$ if $a_{-k} \in OPT$ or 0 otherwise. Then it samples the physical state $s^{(n),t}$ using the state transition function $T_k(S^t|S^{(n),t-1}, a_k^{t-1}, a_{-k}^{t-1})$ (line 3). Then for each possible observation, if the current nesting level $l$ is 1, it calls the 0-level belief update, described in Algorithm 2, to update other agents' beliefs over physical state $b_{-k,0}^t$ (line 5 to 7); or it recursively calls itself at a lower level $l-1$ (line 8 to 11), if $l$ is greater than 1. The sample weights $w_t^{(n)}$ are computed according to observation likelihoods of modeling and modeled agents (line 12, 13). Lastly, the algorithm normalizes the weights (line 15), resamples the intermediate particles(line 16) and resamples another time from similar neighboring models using a Gaussian distribution to avoid divergence (line 17).

---

**Algorithm 2: Level-0 Belief Update**

$b_{k,0}^t = \text{Level0BeliefUpdate}(\theta_{k,0}^{t-1}, a_k^{t-1}, o_k^t)$

1    get $T_k$ and $O_k$ from $\theta_{k,0}^{t-1}$
2    $P(a_{-k}^{t-1}) = 1/|A_{-k}|$
3    for $s^t \in S$:
4        sum=0
5        for $s^{t-1}$:
6            for $a_{-k}^{t-1} \in A_{-k}$:
7                $P(s^t|s^{t-1}, a_k^{t-1}) += T_k(s^t|s^{t-1}, a_k^{t-1}, a_{-k}^{t-1})P(a_{-k}^{t-1})$
8            $sum += P(s^t|s^{t-1}, a_k^{t-1})b_{k,0}^{t-1}(s^{t-1})$
9        for $a_{-k}^{t-1} \in A_{-k}$:
10          $P(o_k^t|s^t, a_k^{t-1}) += O_k(o_k^t|s^t, a_k^{t-1}, a_{-k}^{t-1})P(a_{-k}^{t-1})$
11        $b_{k,0}^t = sum \times P(o_k^t|s^t, a_k^{t-1})$
12    normalize and return $b_{k,0}^t$

---

The 0-level belief update, described in Algorithm 2, takes agent model, $\theta_{k,0}^{t-1}$, action, $a_k^{t-1}$, and observation, $o_k^t$, as input arguments and returns the belief about the physical state, $b_{k,0}^t$. The other agent's actions are treated as noise (line 2), and transition and observation functions are passed in within the first input argument $\theta_{k,0}^{t-1}$. For each possible action $a_{-k}^{t-1}$, it computes the actual state transition (line 7) and observation function (line 10) by marginalizing over others' actions, and returns the normalized belief $b_{k,0}^t$. Notice that the transition and observation functions, $T_k(s^t|s^{t-1}, a_k^{t-1}, a_{-k}^{t-1})$ and $O_k(o_k^t|s^t, a_k^{t-1}, a_{-k}^{t-1})$ contained in $\theta_k^{t-1}$, depend on particular model parameters of the actual agent on the 0th level.

Our interactive belief update algorithm differs in three major ways from the I-PF. First, in order to update the belief over this intentional model space of other agents, their initial belief, transition function, observation function and reward function in their frames are all unknown and become samples. For instance, the set of $n$ samples of other agents' intentional models $\theta_{-k,l-1}^{(n),t-1} = < b_{-k,l-1}^{(n),t-1}, A_{-k}, \Omega_{-k}, T_{-k}^{(n)}, O_{-k}^{(n)}, R_{-k}^{(n)}, OC_k >$. The observation function of the modeled agents, $O_{-k}^{(n)}(o_{-k}^t|s^{(n),t}, a_k^{t-1}, a_{-k}^{t-1})$ in line 12 of Algorithm 1, is now randomized consequently. Second, the transition and observation functions of the level-0 agent, in line 7 and 10 of Algorithm 2, are passed in as input arguments which correspond to each model sample. Lastly, we add another resampling step in line 17 of Algorithm to avoid divergence, by resampling the model samples from a Gaussian distribution with the mean of current sample value. This additional resampling step is nontrivial, since empirically the samples diverge quickly due to the enormously enlarged sample space.

# 4 Experiments

We evaluate our algorithm on the multi-agent tiger problem [7] and UAV reconnaissance problem [2]. The multi-agent tiger game is a generalization of the classic single agent tiger game [11]. It contains additional observations caused by others' actions, and the transition and reward functions involve others' actions as well. The UAV reconnaissance problem contains a 3x3 grid in which the agent (UAV) tries to capture the moving target [2].

## 4.1 Parameterization

The initial step of solving an I-POMDP in our approach is to parameterize other agents' models in terms of an I-POMDP or POMDP, depending on the modeling agent's strategy level. Then, the model parameters can be sampled and updated using the interactive belief update algorithm for solving the planning task.

Here we give an example of parameterization using the tiger problem. The UAV problem has a similar process accordingly. For the simplicity of presentation, assume there are two agent $i$ and $j$ in the game and the strategy level is 1 (but we experiment with higher strategy levels in later sections), then for the two-agent tiger problem: $IS_{i,1} = S \times \theta_{j,0}$, where $S = \{$tiger on the left (TL), tiger on the right (TR)$\}$ and $\theta_{j,0} = < b_j(s), A_j, \Omega_j, T_j, O_j, R_j, OC_j >\}$; $A = A_i \times A_j$ are joint actions of listen (L), open left door (OL) and open right door(OR); $\Omega_i$: $\{$growl from left (GL) or right (GR)$\} \times \{$creak from left (CL), right (CR) or silence (S)$\}$; $T_i = T_j : S \times A_i \times A_j \times S \rightarrow [0,1]$; $O_i : S \times A_i \times A_j \times \Omega_i \rightarrow [0,1]$; $R_i : IS \times A_i \times A_j \rightarrow \mathbb{R}$.

As mentioned before we assume that $A_j$ and $\Omega_j$ are known, and $OC_j$ is infinite horizon with discounting. We want to recover the possible initial belief $b_j^0$ about the physical state, the transition, $T_j$, the observation, $O_j$ and the reward, $R_j$. Thus the main idea of our experiment is to do Bayesian parametric learning with the help of our sampling algorithm.

Table 1: Parameters for transition, observation and reward functions

| S | A | p(TL) | p(TR) |
|---|---|---|---|
| TL | L | $p_{T1}$ | $1 - p_{T1}$ |
| TR | L | $1 - p_{T1}$ | $p_{T1}$ |
| * | OL | $p_{T2}$ | $1 - p_{T2}$ |
| * | OR | $1 - p_{T2}$ | $p_{T2}$ |

| S | A | p(GL) | p(GR) |
|---|---|---|---|
| TL | L | $p_{O1}$ | $1 - p_{O1}$ |
| TR | L | $1 - p_{O1}$ | $p_{O1}$ |
| * | OL | $p_{O2}$ | $1 - p_{O2}$ |
| * | OR | $1 - p_{O2}$ | $p_{O2}$ |

| S | A | R |
|---|---|---|
| * | L | $r_{R1}$ |
| TL | OL | $r_{R2}$ |
| TR | OR | $r_{R2}$ |
| TL | OR | $r_{R3}$ |
| TR | OL | $r_{R3}$ |

We see in Table 1 that it is a large 8-dimensional space to learn from: $b_j^0 \times p_{T1} \times p_{T2} \times p_{O1} \times p_{O2} \times r_{R1} \times r_{R2} \times r_{R3}$, where $\{b_j, p_{T1}, p_{T2}, p_{O1}, p_{O2}\} \in [0,1]^5 \subset \mathbb{R}$ and $\{r_{R1}, r_{R2}, r_{R3}\} \in [-\infty, +\infty]^5$.

Figure 1 illustrates the interactive belief update in the game described above, assuming the sample size is 8. The subscripts denote the corresponding agents and each dot represents a particular belief sample. The propagation step in implemented in lines 2 to 11 in Algorithm 1, the weighting step is in lines 12 to 15, and the resampling step is in lines 16 and 17. The belief update for a particular level-0 model sample, $\theta_j = \langle 0.5, 0.67, 0.5, 0.85, 0.5, -1, -100, 10 \rangle$, is solved using Algorithm 2.

## 4.2 Results

We firstly fix the modeled agent $j$ to be a level-2 I-POMDP agent and experiment with different modeling approaches for agent $i$ in order to compare the performance in terms of average reward. We compare level-3, level-2, level-1 intentional I-POMDP models with a subintentional model, in which agent $j$ is assumed to choose his action according to a fixed but unknown distribution and therefore is called a frequency-based (fictitious play) model [5].

In Figure 2, we see that the intentional I-POMDP approaches has significantly higher reward as agent $i$ perceives more observations, and level-2 I-POMDP performs slightly better than level-1 while level-3 has high variance but at least competes with level-2. The subintentional approach has certain learning ability but is not sophisticated enough to model a rational (level-2 intentional I-POMDP) agent, therefore its performance is worse than all I-POMDP models.

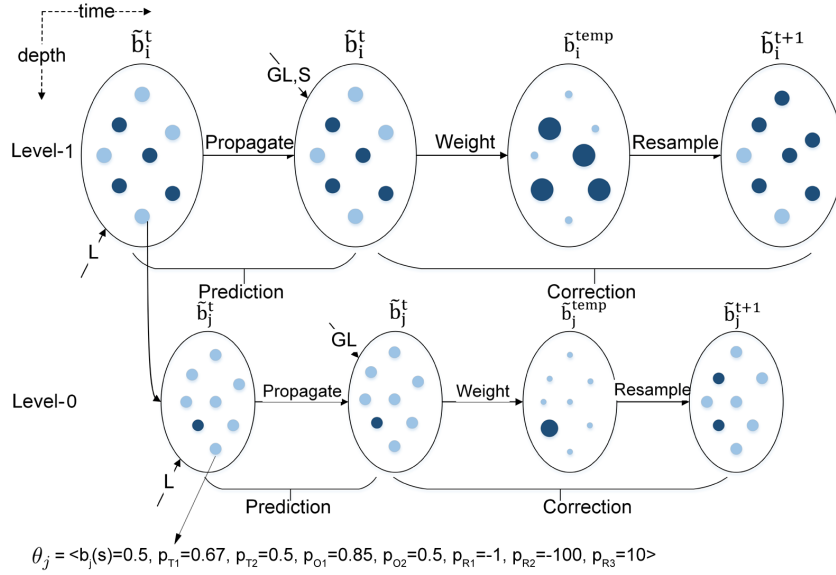

Figure 1: An illustration of interactive belief update algorithm using tiger problem for two-agent and level-1 nesting.

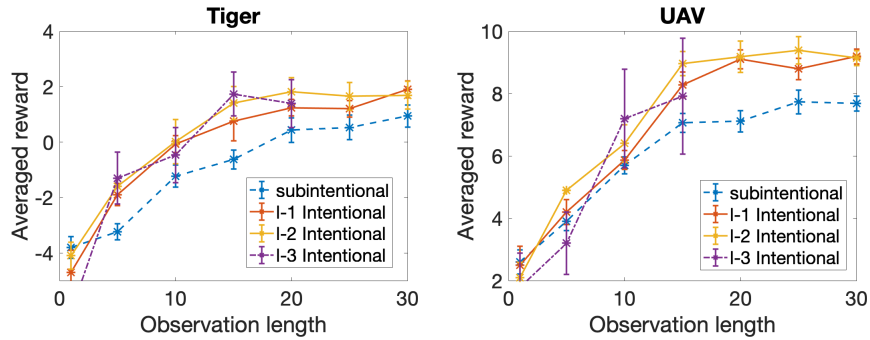

Figure 2: Performance comparison in terms of average reward per time step versus observation length. The plot is averaged on 5 runs and uses 2000 and 1000 samples for tiger and UAV respectively. The vertical bars stand for standard deviations.

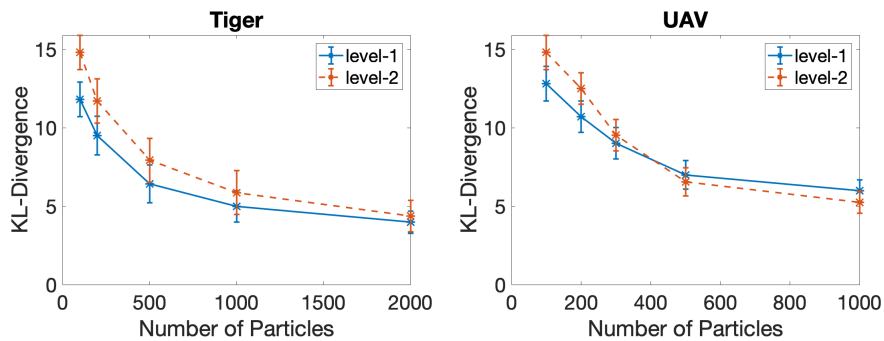

Figure 3: Learning quality, measured by KL-divergence, improves as the number of particles increases. It measures the difference between the ground truth of the model parameters and the learned posterior distributions. The vertical bars are the standard deviations.

In Figure 3 we show that the learning quality, in terms of the sum of independent KL-divergence of each model parameter dimension, becomes better as the number of particles increases. It measures the

difference between the ground truth of the model parameters and the learned posterior distributions by giving the relative entropy of the truth with respect to the posteriors.

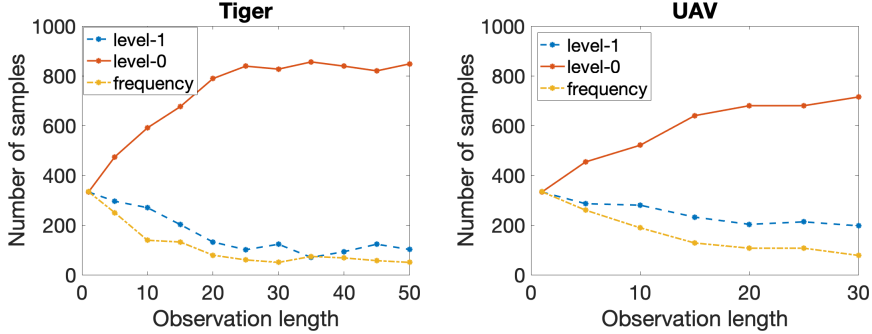

Figure 4: Agent $i$ learns agent $j$'s most likely nesting level. Samples representing $j$'s models of different nesting levels evolve as agent $i$ perceives more observations. Totally 1000 samples are used and experiments start from equal number of level-1, level-0 and frequency-based samples.

Then we fix the modeling agent $i$'s strategy level to be 2 and try to observe the changes of $j$'s samples which represent different possible models or strategy levels. Specifically, we start from equal number of samples that representing $j$ as level-1 I-POMDP, level-0 POMDP, and frequency based agents, and then gradually learn that the majority of samples converge or become close to the ground truth: $j$ is a level-1 I-POMDP.

Table 2: Running time for tiger and UAV problems using various number of samples

| Belief Level | N=500 | N=1000 | N=2000 |
|---|---|---|---|
| 1 | 1.96s ±0.43s | 3.68s ±1.01s | 35.2s ±2.82s |
| 2 | 5m27.23s ±5.19s | 16m36.07s ±10.84s | 49m36.07s (single run) |

Tiger

| Belief Level | N=100 | N=500 | N=1000 |
|---|---|---|---|
| 1 | 4.86s ±1.34ss | 12.31s ±1.39s | 2m1.43s ±3.29s |
| 2 | 2m43.1s ±3.98s | 9m53.7s ±6.48s | 36m19.5s ±18.63s |

UAV

Lastly, we report the running time of our sampling algorithm in Table 2. The computing machine has an Intel Core i5 2GHz, 8GB RAM, and runs macOS 10.13 and MATLAB R2017.

## 5   Conclusions and Future Work

We have described a novel approach to learn other agents' intentional models by making the interactive belief update using Bayesian inference and Monte Carlo sampling methods. We show the correctness of our theoretical framework using the multi-agent tiger and UAV problems in which it accurately learns others' models over the entire intentional model space and can be generalized to problems of larger scale in a straightforward manner. Therefore, it provides a generalized Bayesian learning algorithm for multi-agent sequential decision making problems.

For future research opportunities, the applications presented in this paper can be extended to more complex problems by leveraging emerging deep reinforcement learning (DRL) methods, which already solves POMDPs in an neural analogy [12]. DRL should also be capable of approximating key functions in I-POMDPs, thus has the potential to serve as an efficient computational tool for I-POMDPs.

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
