[Supplementary Material]

# Appendices

## 1 Appendix 1: Tiger Problem Example in Details

Due to limited space, we only give general results in the main paper. In this appendix, we try to illustrate the learning process of our algorithm using the compact tiger problem in more details. For illustrative purpose, we assume there are two agents in the settings, and the modeling agent $i$'s strategy level is 1, which means the other agent $j$ is a level-0 POMDP.

Figure 1: Optimal policies denoted as FSCs of: (a) $\theta_{j_1} = <0.5, 0.67, 0.5, 0.85, 0.5, -1, -100, 10>$, (b) $\theta_{j_2} = \langle 0.5, 1, 0.5, 0.95, 0.5, -1, -10, 10 \rangle$, and (c) $\theta_{j_3} = \langle 0.5, 0.66, 0.5, 0.85, 0.5, 10, -100, 10 \rangle$.

We run experiments with agent $j$ acting according to three different policies shown in Figure 1. In each experiment, we compare the performance of three different modeling agents: a level-1 I-POMDP, a level-2 I-POMDP and a subintentional model (fictitious play). For brevity, we focus on results of learning models of the level-1 agent whose policy is in Figure 1 (a), but give an comparison for all of them in Figure 4.

These three particular opponents are chosen to demonstrate the learning ability of our algorithm. The aim of the first experiment is trying to learn models of agent $j$ who is modeling his opponent $i$ using a no-information model. As shown in Figure 1(a), agent $j$'s actual policy, after normalizing over $i$'s uniform actions, is equivalent to a POMDP policy that looks for three consecutive growls from the same direction and then opens the opposite door. The second experiment involves a POMDP agent $j$ equipped with high listening accuracy of 0.95 and small penalty of -10 for encountering the tiger, i.e. agent $j$ alternately opens doors and listens as shown in Figure 1(b). And the third experiment involves a simple POMDP agent $j$ who always listens since the listening penalty now equals the reward, i.e. 10, as shown in Figure 1(c). One can view the difficulties of learning these three agents' models ($\theta_{j_1}$, $\theta_{j_2}$, and $\theta_{j_3}$) as relatively hard, medium, and easy, since the policy difficulties decrease in these experiments. Meanwhile, more possible values of model parameters can be learned from $\theta_{j_1}$ to $\theta_{j_2}$ to $\theta_{j_3}$, since there are more possible models which can generate the same policy.

For this particular experiment, we want to learn relatively complicated intentional models of agent $j$: $\theta_{j_1} = \langle 0.5, 0.67, 0.5, 0.85, 0.5, -1, -100, 10 \rangle$, which assumes the other agent's (i.e. $i$'s) actions are drawn from a uniform distribution. Equivalently, agent $j$'s actual policy, as shown in Figure 1 (a), is to look for three consecutive growls from the same direction and then open the opposite door. For this particular experiment, we assume $i$ always listens, and we simulate $i$'s observation history for

Figure 2: Histograms of sample values of assigned uniform priors (top row) and learned posterior distributions (bottom raw) over parameters of model $\theta_{j1} = \langle 0.5, 0.67, 0.5, 0.85, 0.5, -1, -100, 10 \rangle$ in Figure 1 (a). The modes of the posteriors are close to the true model parameters.

the sake of firstly verifying the correctness of our algorithm, excluding the impact of uncertainties in the transition function and of the hearing ability. The simulated observation history is as follows, which are exactly three growls from the same direction and then a creak caused by $j$'s door-opening action: {GL,S GL,S GL,S GL,CR GL,S GL,S GL,S GR,CR GL,S GL,S GL,S GR,CR GL,S GL,S GL,S GR,CR GR,S GR,S GR,S GR,S GR,CL GR,S GR,S GR,S GR,CL GR,S GR,S GR,S GR,CL GR,S GR,S GR,S GR,CL GR,S GR,S GR,CL GR,S GR,S GR,S GR,CL GR,S GR,S GR,S GL,CL GL,S GL,S GL,S GR,CR GL,S GL,S GL,S GR,CR GR,S GR,S}

To exclude the potential impacts from informative prior belief distributions, we assign uninformative uniform priors to each parameter, samples of which are shown in the top row of Figure 1. These uniform priors are: $\{b_j^0, p_{T1}, p_{T2}, p_{O1}, p_{O2}\} \sim U(0,1)^5$, $p_{R1}, p_{R2}, p_{R3} \sim U(-200, 200)^3$. After 50 time steps, the samples converge to posterior distributions over agent $j$'s model parameters, the results are given in the bottom row of Figure 2. Since the parameter space is 8-dimensional, here we show the marginal distributions of each parameter in histograms. We can see that most of the samples are centered around the true parameter values. We give the KL-divergence plot in Figure 3.

Figure 3: Learning quality measured by KL-divergence improves as the number of particles increases. It measures the difference between the ground truth of the model parameters and the learned parameters, as defined in Equation 1 and 2. The vertical bars are the standard deviations. Fixed number of bins (50) are used to compute the discrete probabilities.

In Figure 3 we show that the learning quality of these three experiments in terms of the KL-divergence. It measures the difference between the ground truth of the model parameters and the learned posterior distributions by giving the relative entropy of the truth with respect to the posteriors. We define the KL-divergence as the sum of independent KL-divergence of each model parameter dimension, as shown in Equation 1.

$$D_{kl}[b(\theta)||\tilde{b}(\theta)] = \sum_{d=1}^{D} D_{kl}[b(\theta_d)||\tilde{b}(\theta_d)] \qquad (1)$$

where $b(\theta)$ denotes the true model parameters and $\tilde{b}(\theta)$ denotes the sampled posterior distribution, $\theta$ is a model represented by 8D parameters, and $d = 1:D$ is the parameter dimension.

For each parameter dimension $d$, the KL-divergence $D_{kl}[b(\theta_d)||\tilde{b}(\theta_d)]$ reduces to $-\log[\tilde{b}(\theta_d \in Bin_{truth})]$ as shown in Equation 2.

$$\begin{aligned} D_{kl}[b(\theta_d)||\tilde{b}(\theta_d)] &= -\sum_{bin} b(\theta_d)\log(\frac{\tilde{b}(\theta_d)}{b(\theta_d)}) \\ &= -\log[\tilde{b}(\theta_d \in Bin_{truth})] \\ &= -\log[\frac{count(\theta_d \in Bin_{truth})}{N}] \end{aligned} \qquad (2)$$

since

$$b(\theta_d) = \begin{cases} 1 & \text{, when } \theta_d \in Bin_{truth}, \\ 0 & \text{, otherwise.} \end{cases} \qquad (3)$$

where $Bin_{truth}$ is the bin containing the ground truth of the model parameter, $N$ is the total number of samples.

Figure 4: Performance comparisons in terms of prediction error rate vs observation length for $\theta_{j1} = \langle 0.5, 0.67, 0.5, 0.85, 0.5, -1, -100, 10 \rangle$

Because agent $i$ is now able to learn $j$'s likely models, it should be capable of predicting $j$'s actions relatively accurately. Therefore, we tested the performance of our algorithm in terms of prediction accuracy towards $j$'s actions. For conciseness, we show the average prediction error rates for all three experiments in Figure 4. We compared the results with other modeling approaches, such as a frequency-based (fictitious play) approach, in which agent $j$ is assumed to choose his action according to a fixed but unknown distribution, and a no-information model which treats $j$'s actions as uniform noise. The shown results are averaged plots of 10 random runs, each of which has 50, 30 and 30 time steps respectively. It shows that the intentional I-POMDP approach has significantly lower error rates as agent $i$ perceives more observations. The no-information model assumes $j$'s actions are drawn from a uniform distribution, therefore has a fixed high error rate. The frequency based approach has certain learning ability but is not sophisticated enough to be able to model a rational agent, therefore its performance falls in between the other two.

In Figure 5, we show a brief demonstration of learning the first model of agent $j$: $\theta_{j_1}$. Since the original parameter space is 8-dimensional, we use the principal component analysis (PCA) to reduce it to 2D and plot it out as a 3D histogram, as shown in Figure 5. We see that model samples in the cluster gradually concentrate to the center, which is the true model: $\theta_{j_1} = \langle 0.5, 0.67, 0.5, 0.85, 0.5, -1, -100, 10 \rangle$. At the moment, the mean value of this cluster, $\tilde{\theta}_{j_1} = \langle 0.49, 0.69, 0.49, 0.82, 0.51, -0.95, -99.23, 10.09 \rangle$, is very close to the actual model.

Figure 5: Histograms of all model samples at various time steps during learning, after projecting from 8D to 2D, show that samples are gradually concentrating to the center of the cluster which represents the true model.