[Reviews · NeurIPS 2018]

Reviewer 1



Main idea: This paper applies Monte-Carlo sampling to interactive POMDPs. In interactive POMDPs agents’ belief over the state includes a belief over the belief of other agents. In particular the authors show that intentional models produces better results on two standard POMDP benchmarks (Tiger and UAV). An intentional models other agents are as pursuing goals rather modeling their behaviour. Comment: This paper is currently hard to follow unless the reader is already familiar with the I-POMDP framework. Results: The results seem rather inconclusive. There are no standard baselines provided for comparison. The only takeaway is that the intentional models (of any level) do better than the subintentional one. No p-values are stated for the results. It is surprising that the results are not compared to standard multi-agent RL methods for addressing partially observable settings (this is left for future work). Algorithm 2: ‘sum’ variable is never initialised. Line 9: P(o) + O * P: This line has no effect. Possibly a typo? These two issues make the interpretation of the algorithm difficult. Minor issues: Line 183: “agent’s strategy level. And then the model” [run-on sentence] Line 165: “belief over intentional model space” -> “belief over the intentional model space” (‘the’ is missing in more places in the text, probably good to run a grammar checker). I found the rebuttal and other reviews insightful and my concerns mostly addressed. I have improved my score to reflect this.

Reviewer 2



The paper describes a sampling method for learning agent behaviors in interactive POMDPs (I-POMDPs). In general, I-POMDPs are a multi-agent POMDP model which, in addition to a belief about the environment state, the belief space includes nested recursive beliefs about the other agents' models. I-POMDP solutions, including the one proposed in the paper, largely approximate using a finite depth with either intentional models of others (e.g., their nested beliefs, state transitions, optimality criterion, etc.) or subintentional models of others (e.g., essentially "summaries of behavior" such as fictitious play). The proposed approach uses samples of the other agent at a particular depth to compute its values and policy. Related work on an interactive particle filter assumed the full frame was known (b, S, A, Omega, T, R, OC). The main unique characteristic of the proposed approach is to relax this assumption: only S, A, Omega are known, with beliefs, T, O, and R learned from n samples. The algorithm essentially proceeds by sampling from these distributions instead to generate beliefs and thus compute value and policy. Experiments were conducted on the multi-agent tiger and UAV domains, comparing a fictitious play I-POMDP baseline versus 0-, 1-, 2-, 3-level intentional I-POMDPs. In summary, results show that the intentional outperforms the baseline. The paper does a great job of explaining the I-POMDP and the issues surrounding the nested modeling of other agents' models. While the algorithm suffers from the necessary notational overhead, each step is well explained in the surrounding companion paragraphs. As there are no theoretical results, experiments attempt to serve as marginally convincing evidence that the approach works. Here are some comments and questions: 1. The experiments require up to 1000 or 2000 samples to obtain solid performance to accurately model T, O, R, etc. These problems are quite small, with simple transition and observation functions to learn. Given this fact, and the long run times for basic 2-level nesting, what is the main motivation for the usefulness of this approach? Why not use a subintentional model for any larger I-POMDPs? The number of samples in this approach will need to grow rapidly to astronomical sizes, with likely large computation times. Perhaps a stronger motivation to contrast this with the much less sample-dependent---but knowingly lower performant---subintentional models. 2. Experiments show interesting behavior regarding the performance, number of samples, and even KL-divergence between learned model parameters and ground truth. However, in application, how do we know how many samples are sufficient to accurately model the problem? Are there any techniques for determining a stopping point, especially in larger I-POMDPs when there are many more samples? Overall, the idea clearly is how one would solve the problem of not having T, O, and R for other agents a priori: Monte Carlo sample. The paper partially delivers on the promise of such a scalable algorithm, for which simple subintentional models like fictitious play already serve this role well. That said, the idea is good and broadly moving in this model-based direction is well-motivated. Some minor comments: - Line 208: "...with an subintentional..." -> "...with a subintentional...". - Line 219: "Then we fixe the..." -> "Then we fixed the..." - There are a lot of figures, many of which could be saved for the supplement in favor of some theoretical result, perhaps regarding number of samples (as above), error bound, etc. - References have a few typographic errors. "POMDP" needs to be capitalized, "Markov", "Bayesian", etc.

Reviewer 3



This work extends the interactive particle filter to solve I-POMDP where the intentional model of the other agents (i.e., the transition, observation and reward functions) are unknown as a prior. Hence, when making decision, each agent must learn the other agents’ belief about the state, the transition, the observation, and the reward function. With initial belief samples, the proposed algorithm starts from propagating each sample forward in time and computing their weights, then it resamples according to the weights and similarity between models. This is very similar to Bayesian learning for POMDP but for multi-agent cases modeled as I-POMDPs. This work is evaluated mainly empirically. It is not clear how the reward function can be learned by sampling because there is no signal about the reward function in the observation as in the standard I-POMDPs. Update: Thanks for answering my question. I understand the reward function is part of the sampled model.